# The Burden of Alcohol-Related Emergency Department Visits in a Hospital of a Large European City

**DOI:** 10.3390/healthcare11060786

**Published:** 2023-03-07

**Authors:** Hanna Cholerzyńska, Wiktoria Zasada, Tomasz Kłosiewicz, Patryk Konieczka, Mateusz Mazur

**Affiliations:** 1Students’ Scientific Circle of Emergency Medicine, Poznan University of Medical Sciences, 7 Rokietnicka Street, 60-608 Poznań, Poland; 2Department of Medical Rescue, Poznan University of Medical Sciences, 7 Rokietnicka Street, 60-608 Poznań, Poland; 3Department of Emergency Medicine, Poznan University of Medical Sciences, 7 Rokietnicka Street, 60-608 Poznań, Poland

**Keywords:** alcohol-related admissions, emergency department, alcohol intoxication, alcohol-related trauma, admission rates

## Abstract

(1) Alcohol consumption contributes to the development of numerous diseases and is a big organizational burden on emergency departments (EDs). (2) We examined data on alcohol-related ED admissions in Poznan, Poland between 1 April 2019 and 31 March 2022. A total of 2290 patients’ records were collected and analysed. The main goal was to determine the impact that these visits had on the functioning of the ED and the hospital. (3) The alcohol-related admission rate was significantly higher in males (78.95% vs. 21.05%), and the median blood alcohol concentration (BAC) level was 2.60 (1.78–3.38) ‰. Most of the visits took place at weekends and in the evening. Patients with higher BAC tended to stay longer in the ED, but had a lower chance of being admitted. A majority of patients required radiology and laboratory testing, 20.44% needed psychiatric examination, and 19.69% suffered trauma, mainly to the head. (4) Injuries and mental problems were the most common medical emergencies. This study presents trends in alcohol-related ED attendances, examines reasons for visits, and makes an attempt to assess overall burden on EDs.

## 1. Introduction

Consumption of alcohol has always been a huge part of Polish culture and tradition. According to World Health Organization (WHO) data, mean ingestion of pure alcohol per capita in Poland was 10.96 L in 2019, 10.62 L in 2018, and 10.53 L in 2017, with a noticeably increasing tendency [1,2]. The State Agency for the Prevention of Alcohol-Related Problems provides similar information: an increasing alcohol consumption in Poland from 1993 to 2019 (9.45, 9.55, 9.78 L of pure alcohol per capita in the last 3 years) [3]. According to the WHO’s alcohol country fact sheet for Poland, in 2008 total alcohol per capita consumption started to exceed the European Union average and has remained higher ever since [1]. In 2020, the prevalence of heavy episodic drinking (meaning consumption of at least 60 g or more of pure alcohol on at least one occasion every month) was higher in Poland than the European mean—15.8 and 14.8, respectively [4].

Alcohol consumption is not only associated with many health conditions, such as heart failure, hypertension, and hepatic cirrhosis, but also with an increased risk of car crashes and violence [5,6,7,8]. All these factors may result in a patient needing immediate medical help.

In Poland, alcohol-attributable fractions were responsible for 8.2% and 2.9% of all cancer deaths (in males and females, respectively) and 48.1% of car crash deaths [9,10]. Furthermore, it was estimated that alcohol was responsible for 3.8% of all global deaths, 4.6% of global disability-adjusted life-years, and 1.3–3.3% of total health costs [11,12].

Emergency departments (EDs) serve multiple functions in the healthcare system, linking primary health hospitals, ambulance systems, and disaster response teams [13,14]. Thus, they are also the first place to seek medical help for patients with alcohol-related problems. The relationship between alcohol consumption and ED visits has been studied repeatedly in numerous countries. The rate of alcohol-related attendance was estimated to be from 1.2% in Belgium to 28% in the USA [15,16]. Appropriate alcohol patient management seems to be crucial to effective public health service utilisation.

The purpose of this study was to explore trends in alcohol-related ED attendances, examine reasons for visits, and to make an attempt to establish overall burden on EDs (such as use of treatment and diagnostic resources, length of stay) in order to improve management patterns and appropriate hospital resources utilisation.

## 2. Materials and Methods

### 2.1. Study Design

We retrospectively analysed visits between 1 April 2019 and 31 March 2022 (3 full years of data collection) and searched for the words “alcohol” or “intoxication” in medical records. Each record was then manually checked by the authors.

### 2.2. Legal Issues

According to Polish law, retrospective analysis of medical records does not require approval of the Bioethics Committee at Poznan University of Medical Sciences (confirmation KB 701/22).

### 2.3. Study Setting

The study was conducted in the ED at Hipolit Cegielski Medical Centre in Poznan, Poland. It is an ED visited by approximately 32,500 patients every year, predominantly adults. Data were based on information captured systematically using the hospital’s software.

### 2.4. Participants

The study population included patients treated in the Emergency Department. Our inclusion criteria were:1.age 18 years and above;2.confirmed alcohol consumption or positive alcohol test result, with presence of suspected alcohol consumption based on clinical symptoms (slurred speech, impaired balance, aggression, increase in talkativeness, altered perception of the environment); and3.positive alcohol test in patients who were unconscious or had altered mental status.4.In cases of doubt about a record’s inclusion, a decision was made after two authors were independently consulted.

### 2.5. Data Sources and Measurement

The collected data included age and sex, ICD-10 (*International Classification of Diseases*—10th revision) diagnosis, timing of the ED visit, including total length of stay in the ED (defined as the time between examination made by ED physician and final decision (admission/discharge)), day of week, and time of day: night (23:00–6:59), day (07:00–14:59), and afternoon and evening (15:00–22:59). Data were also filtered for procedures performed on patients, including any medication administration with sedatives filtered separately, wound sutures, limb immobilizations and radiological scans (both plain X-rays and computerized tomography (CT) scans). We also determined the consequent location of the patient after ED visit (whether it was discharge, transfer to another ward, or death in the ED). All the data were exported to data sheets, filtered, and checked for flaws.

### 2.6. Variables

#### 2.6.1. Primary Outcomes

Our primary outcomes were defined as: distribution of admissions depending on the year, month, and day; length of stay (LOS—defined as time between first medical examination made by emergency physician and final decision—transfer to another ward or discharge); and resources needed to accurately diagnose and treat patients. Resources were defined as need for X-ray or CT scan, administration of medications, any laboratory tests, sedation, suturing, decontamination, and continuous monitoring of vital signs.

#### 2.6.2. Secondary Outcomes

Our secondary outcomes were focused on recognizing major medical problems. These were defined according to the ICD-10.

### 2.7. Statistical Methods

First, data were extracted from the hospital information system and prepared in a database created in Microsoft Excel (Microsoft Corporation, Redmond, WA, USA). Then, analysis was performed using Statistica 12 software (Tibco Inc., Tulsa, OK, USA). Descriptive statistics of measurable variables were performed. The categorical variables were expressed as numbers (*n*) with percentages (%) and quantitative data as medians (interquartile range), as they did not present normal distribution (confirmed by Wilk–Shapiro W test). To evaluate the relationship between blood alcohol concentration (BAC) and LOS, linear regression with Pearson’s linear correlation coefficient was used as appropriate. To perform the multivariate analysis, we used two models. A multivariate regression was used for analysis of the association between quantitative variables (BAC and age) and LOS. A two-way ANOVA test was used for multivariate analysis of categorical variables. For this purpose, the group was divided according to BAC (‰) (<1; 1–2; 2–3; 3–4; >4). The strength of the effect (eta^2^) was given only for statistically significant relationships. A value of *p* < 0.05 was considered statistically significant.

## 3. Results

There were 3174 patients in the analysed period of time, and 884 records were excluded due to lack of sufficient data (brought by the police to be examined before custody, left without being seen by a physician, missing data, or presented themselves in order to enter drug or alcohol rehabilitation). Those visits did not require any resources, except physical examination and often only administrative work. Therefore, the study group consisted of 2290 patients. Most of them were male (*n* = 1808, 78.95%). The number of all patients in the hospital during the study was 97,233. Alcohol-related ED attendances accounted for 3.26% of all visits.

Median age was 41 (30–54) years. The youngest patient was 18 and the oldest 90.

Median BAC was 2.60 [1.78–3.38] ‰. The lowest BAC was 0.21‰ and the highest 6.42‰.

Most patients were referred to the ED by paramedics (69.00%, *n* = 1580). The rest (31.00%, *n* = 710) walked in without previous medical assistance.

### 3.1. Primary Outcomes

#### 3.1.1. Distribution of ED Visits

Mean number of visits in the ED was 190 ± 22 patients per month. The highest number was admitted in December (*n* = 225), the lowest in March (*n* = 159).

When related to the time of attendance, most individuals presented at 15:00–22:59 (44.10%, *n* = 1011), then at night (23:00–6:59; 32.85%, *n* = 753) and in the morning (07:00–14:59; 23.03%, *n* = 528).

We found that most patients were admitted at weekends—Saturday (18.61%, *n* = 427) and Sunday (17.20%, *n* = 394)—followed by Friday (14.23%, *n* = 326), Wednesday (13.10%, *n* = 300), Tuesday (12.35%, *n* = 283), Thursday (12.27%, *n* = 281) and Monday (12.18%, *n* = 279).

Triage categories were red (3.10%, *n* = 71), orange (24.32%, *n* = 557), yellow (58.08%, *n* = 1330), green (13.62%, *n* = 312), and blue (0.87%, *n* = 20).

Median LOS was 233 (137–384) min. The shortest LOS was 3 min, whereas the longest was 1405 min. The LOS in the ED was related to the BAC. Patients with a higher BAC stayed longer in the ED (Figure 1).

A total of 619 admissions were made, which accounted for 27.04% of all alcohol-related visits. The Mental Health Facility (MHF) admitted 399 patients (64.46% of all admissions). This was followed by the Department of General Surgery—63 patients (10.17% of all admissions), Intensive Care Unit—25 patients (4.03% of all admissions), Department of Internal Medicine—25 patients (4.03% of all admissions), Department of Orthopaedic Surgery—20 patients (3.32% of all admissions), Department of Neurology—15 patients (2.42% of all admissions), and Department of Neurosurgery—7 patients (1.13% of all admissions). Admissions to other wards, such as Maxillofacial Surgery, Toxicology, Infectious Diseases, and Vascular Surgery accounted for less than 1% of all admissions each.

The highest admission rate was in the group of patients with BAC < 1.0‰ and the lowest in the group of patients with BAC ≥ 5.0‰. Details are presented in Table 1.

Linear regression analysis revealed a weak but statistically significant relationship between BAC and LOS (r = 0.1721, *p* = 0.0000). The relationship is presented in Figure 2.

The multivariate regression result showed a significant effect of BAC on LOS (*p* < 0.001) and no significant effect of age on LOS (*p* = 0.2047). The association between the variables was also statistically insignificant (R^2^ = 0.0303).

The results of the multivariate ANOVA analysis showed that LOS was influenced by BAC (*p* < 0.001), drug administration (*p* < 0.001) decontamination process (*p* < 0.001) and sedation (*p* < 0.001). However, the interaction was statistically significant only for BAC and medication. This means that BAC may be considered an independent factor of prolonged LOS. Details of the multivariate analysis are presented in Table 2.

#### 3.1.2. Resources

Seventy-nine patients (3.45%) required no resources during their stay in the ED. One resource was required by 198 patients (8.65%), two resources by 575 patients (25.12%), three resources by 979 patients (42.76%), four resources by 421 patients (19.39%), and five resources by 37 patients (1.62%). Table 3 summarises the most common resources required by patients under the influence of alcohol.

### 3.2. Secondary Outcomes

A total of 277 different diagnoses were made by ED physicians. The most common diagnoses were: mental and behavioural disorders due to use of alcohol (F10) and toxic effect of alcohol (T51), followed by various head injuries. The 10 most common diagnoses are summarised in Table 4. Together, they constituted 66.57% of all cases. Each of the other diagnoses accounted for less than 1% of all cases.

Interviews from patients or paramedics revealed that 26.64% (*n* = 610) of patients had experienced some sort of trauma. However, during the medical examination, injuries were found in 19.69% (*n* = 451) of individuals. Specific localization of injuries is presented in Table 5.

We also found that 20.44% of our patients (*n* = 468) experienced suicidal thoughts or attempts. Almost half of them (10.61 of total, *n* = 234) were transferred from the MHF due to alcohol intoxication. In 3.62% of individuals (*n* = 83), additional drug abuse was found.

## 4. Discussion

The aim of this study was to analyse data on alcohol-related ED attendances acquired from a hospital in a large European city and establish potential consequences for the healthcare sector. Many studies have covered this topic [17]. However, most of them did not emphasise the overall burden that alcohol-related presentations cause.

The burden of alcohol-related attendances extends beyond EDs, from the involvement of paramedics and Emergency Medical Services (EMS) and continuing with ward admissions and further outpatient consultations. Joseph et al. showed that 30% of EMS calls had alcohol as a contributing factor [18]. Several studies covering alcohol-related ED visits described this variable, but a majority of patients were transported by paramedics, which is consistent with our findings [19,20,21]. Pirmohamed et al. found that alcohol-related problems accounted for 12% of all ED attendances, 50% were aged 18–39 years, and acute alcohol intoxication was the commonest presenting complaint. Moreover, overall, 6.2% of all hospital admissions were due to alcohol-related problems [22]. These findings show what a major impact alcohol may have on the emergency system. In our study, the percentage of alcohol-related attendances was lower, but this could be due to the much longer study period. Verelst et al. during their 12-month study reported that ED visits due to alcohol accounted for 1.2%, which is consistent with our findings [15]. Gender distribution showed the number of males was more than triple that of females. This was an expected tendency, but in earlier studies the difference was smaller [18,21]. This can reflect higher popularity of alcohol consumption in males as well as double the prevalence of heavy episodic drinking in males and ninefold the prevalence of harmful alcohol use in males [23,24].

In our population, the highest admission rate was in December. This tendency can add to the overall burden of alcohol-related ED attendances because of increased staff absence during Christmas/New Year’s Eve. In a study by Knudsen et al., in December, alcohol intake was reported to be the lowest; however, registered alcohol sales were the highest [25]. We found a strong prevalence in admissions at weekends in the afternoon and during the night. This time of decline in performance overlaps with the time of increased number of total attendances, which can worsen the challenges that medical staff must overcome. Results from Belgium confirm similar distribution of visits [15].

Another variable that adds to the overall burden is LOS. Patients with higher BAC stayed in ED longer. Previous papers provided different information. In one study, there was a positive correlation between BAC and LOS [26]. By contrast, Klein et al. in a multicentre study from six EDs found no correlation [27]. However, they excluded patients with alcohol concentration lower than 80 mg/dl, repeat visits and ED admissions resulting in hospital admission. Their study design assumed that only patients impaired from alcohol intoxication and without other comorbidities (that may result in hospital admission) should be counted. Our study included all patients under the influence of alcohol, similarly to the study by Verels et al. [15] The fact that a patient has another medical process ongoing does not explain why they are presenting drunk to the ED. It was also postulated by Klein et al. that if the discharge criterion is “clinical sobriety,” then patients with higher tolerance may be discharged sooner [27]. Patients without other serious medical conditions are left in the ED to sober up. Thus, the bed is taken and ED workers are obligated to provide continuous care. An additional factor is that patients requiring help from the MHF (suicide, psychosis, addiction) are not admitted if they are drunk, so they are sent back to the ED to sober up. This is an internal hospital policy, but it adds to the burden. In Poland, there are no clear guidelines on whether and when a patient under the influence of alcohol can be discharged. This issue is highly facility-dependent and physician-dependent. It is commonly accepted as clinical practice to discharge a patient who is able to walk unassisted, take fluids and without impaired consciousness. Moreover, linear regression analysis revealed a weak but significant relationship between BAC and LOS. Additional important information is that even patients with lower BAC stayed in ED for at least 3 h and statistical outliers reach 20 h. This finding draws attention to the fact that even patients with lower BAC stay long in the ED and affect its work. Multivariate analysis confirmed this thesis and also added information that medication administration significantly prolonged LOS.

Another important finding was an inverse relationship between BAC and possibility of ward admission. Higher BAC can result in symptoms of alcohol intoxication, which can be the reason to present to ED, such as ataxia, vomiting, cardiac arrhythmia, and coma [28]. However, alcohol intoxication itself is not a condition that requires longer hospitalisation and can be successfully managed in the ED. Alcohol intoxication-related situations that may require hospitalisation include hepatic encephalopathy, Wernicke’s encephalopathy, cardiac and respiratory collapse, and severe hyponatraemia [28]. Patients with those conditions rarely present to the ED, as it usually requires prolonged and heavy alcohol consumption. Lower BAC is less likely to present with symptoms requiring medical assistance due to alcohol intoxication, which can implicate that those patients have additional conditions and those conditions can require ward admission. We found multiple studies that described an impact of positive BAC on increased risk of various conditions [29,30,31,32]; however, we did not find any study that described a relationship between level of BAC and probability of ward admission.

Distribution of admissions was similar to that established in previous papers with predominance of trauma-related wards (general surgery, orthopaedic surgery) and ICU [33]. However, in our study the overwhelming majority of patients were admitted to the MHF. This also reflects the distribution of major presentations established using ICD-10 codes. Alcohol-related disorders and toxic effects of alcohol contributed to 40% of all diagnoses, followed by various injuries. This finding underlines the importance of the impact of alcohol on mental health and draws attention to alcohol-related presentations that are not associated with any trauma or directly alcohol-related diseases (e.g., liver disease). Moreover, we found that many patients experienced suicidal thoughts/attempts, which is in line with previous studies that established alcohol as a predictor of suicide attempts [34]. This finding can also contribute to a high number of patients admitted to the MHF. Although psychiatric hospitalization remains the typical disposition for patients in acute crisis with moderate to high suicide risk, some patients at low risk may be discharged to supportive, stable environments [35].

We also assessed the demand for interventions that engaged staff or resources. The more resources a patient requires, the more burden they are causing. The need for two or more resources was previously associated with higher rates of hospital admission and increased LOS [36]. The most frequently used resource was radiology. This can be explained by various injuries, most commonly to the head, followed by extremities, multiple sites, and trunk. Almost 90% of patients required two or more resources. Comparing our results to available data on non-alcohol-related ED visits, we noticed that the use of resources was much higher in our study, e.g., in a study by Müller et al. on non-alcohol-related ED visits, 19.2% of patients required a radiology procedure (X-ray, CT scan) vs. 95.46%, and 16.2% required laboratory workup vs. 82.75% in our study [37]. Verelst et al. reported that 74.3% of alcohol-related ED patients required some form of treatment (fluid, vitamin B_1_, drug administration), 83.1% had laboratory tests done, 23.2% X-ray, and 12.3% CT scan (35.5% for both), which is in line with our findings [15]. This comparison underlines our thesis that alcohol-related ED attendances are a huge burden on ED functioning and the healthcare system.

Another important burden on the healthcare system is the cost of treating patients. According to the public insurance system in Poland, patients are not charged anything unless they lack insurance. Using the average cost of treating a patient from ED presentation to ED discharge estimated by Verelst et al. to be EUR541.32 per patient and mean number of patients per month, average cost of alcohol-related ED attendances is EUR102,850.80 per month [15]. It is an enormous sum, and thus educational and preventative measures should be encouraged and implemented by all healthcare workers.

The authors are aware of several limitations of this study. In the city where the study was conducted, there are also EDs in other hospitals. Therefore, the study group was limited to only one area of the city. Alcohol consumption depends on many factors and is sometimes highly variable from one region to another. This variable was not analysed in this study. Conclusions should therefore be transferred to other groups with great caution. In addition, our hospital has the only MHF working 24/7 in the city, which may have influenced the higher number of mental health patients admitted. Furthermore, not every patient in our ED was tested for alcohol. Moreover, a large part of our patients’ population was excluded from the study group. Their presence does not increase the amount of resource use and work of medical staff, but can add to administrational burden on ED functioning; however, we are aware of this being a limitation of our study. In all probability, we overlooked those in the study who had consumed a sufficiently minor amount of alcohol that they did not present clinical signs of alcohol consumption.

## 5. Conclusions

Alcohol-related ED admissions regardless of BAC levels are a huge burden on the ED. Head injuries and mental problems were the most common medical emergencies. We advise an increase in education and other preventative measures, as well as improvement in accessibility of outpatient mental health and alcohol counselling. Clear guidelines should also be developed for the management of patients under the influence of alcohol with suicidal thoughts.

## Figures and Tables

**Figure 1 healthcare-11-00786-f001:**
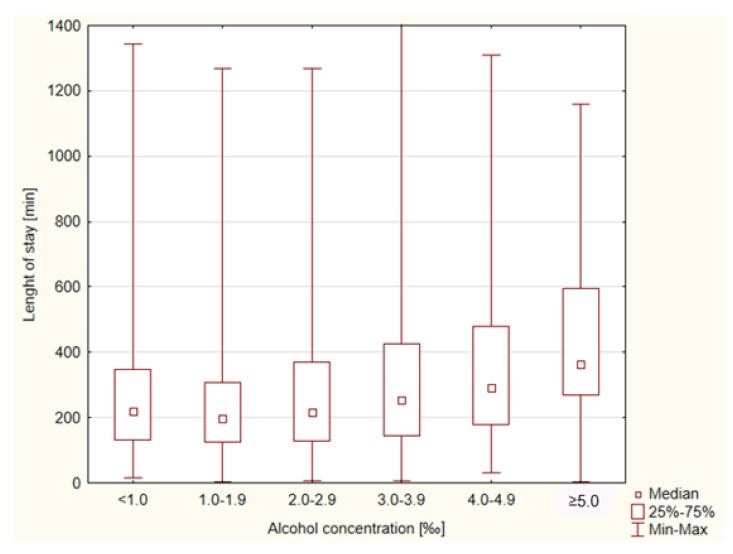
Median duration of patients’ stay in ED (measured in minutes) according to their measured blood alcohol concentration (BAC) 3.1.2. Admission Rate.

**Figure 2 healthcare-11-00786-f002:**
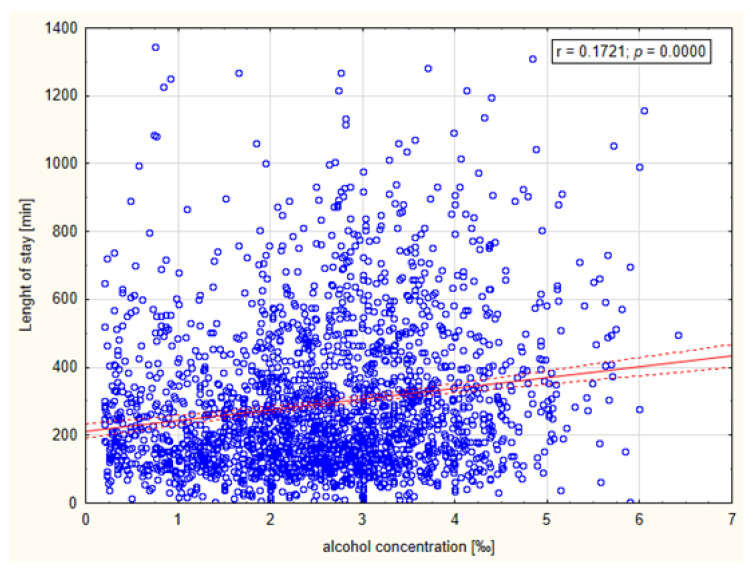
Relationship between patients’ blood alcohol concentration and their length of stay in ED (measured in minutes).

**Table 1 healthcare-11-00786-t001:** Admission rate of patients to ED depending on their blood alcohol concentration.

BAC ^1^ (‰)	Admission Rate (%)
<1.0	38.40
1.0–1.9	38.34
2.0–2.9	28.41
3.0–3.9	17.70
4.0–4.9	15.76
≥5.0	5.77

^1^ BAC—blood alcohol concentration.

**Table 2 healthcare-11-00786-t002:** The results of multivariate analysis concerning BAC and other covariates and its impact on the length of stay (LOS).

Source	SS	DF	MS	F	*p*	Eta-Squared
BAC ^1^	3,999,189	4	999,797	21.746	<0.001 *	0.0367
sex	16,734	1	16,734	0.364	0.5463	
interaction	802,220	4	200,555	4.362	0.0016 *	0.0076
Error	104,732,907	2278	45,976			
BAC ^1^	832,428	4	208,107	4.5086	0.0012 *	0.0078
radiology	34,323	1	34,323	0.7436	0.3885	
interaction	388,466	4	97,117	2.1040	0.0778	
Error	105,146,483	2278	46,157			
BAC ^1^	1,244,006	4	311,002	6.730	<0.001 *	0.0116
laboratory	93,230	1	93,231	2.017	0.1556	
interaction	142,849	4	35,712	0.773	0.5427	
Error	105,269,567	2278	46,211			
BAC ^1^	2,937,682	4	734,421	16.406	<0.001 *	0.0280
medications	2,621,366	1	2,621,366	58.558	<0.001 *	0.0250
interaction	436,985	4	109,246	2.440	0.0449 *	0.0042
Error	101,974,601	2278	44,765			
BAC ^1^	3,117,824	4	779,456	17.356	<0.001 *	0.0295
sedation	2,977,058	1	2,977,058	66.288	<0.001 *	0.0282
interaction	298,316	4	74,579	1.661	0.1564	
Error	102,306,551	2278	44,911			
BAC ^1^	850,714	4	212,679	4.6083	0.0010 *	0.008
suturing	18,885	1	18,885	0.4092	0.5224	
interaction	378,143	4	94,536	2.0484	0.0851	
Error	105,131,932	2278	46,151			
BAC ^1^	167,389	4	41,847	0.9097	0.4572	
decontamination	411,859	1	411,860	8.9537	0.0027 *	0.0039
interaction	316,435	4	79,109	1.7198	0.1428	
Error	104,784,962	2278	45,999			
BAC ^1^	3,186,348	4	796,587	17.267	<0.001 *	0.0297
day of the week	375,109	6	62,518	1.355	0.2292	
interaction	1,224,723	24	51,030	1.106	0.3273	
Error	103,941,645	2253	46,135			
BAC ^1^	3,255,132	4	813,783	17.659	<0.001 *	0.0307
month	723,288	11	65,753	1.427	0.1537	
interaction	1,957,041	44	44,478	0.965	0.5375	
Error	102,671,258	2228	46,082			
BAC ^1^	2,706,758	4	676,689	14.651	<0.001 *	0.0251
Time interval	146,408	2	73,204	1.585	0.2051	
interaction	469,089	8	58,636	1.270	0.2548	
Error	104,983,332	2273	46,187			

^1^ BAC—blood alcohol concentration. Statistically significant findings are marked with asterisks (*). The strength of the effect (eta^2^) is given only for statistically significant relationships.

**Table 3 healthcare-11-00786-t003:** Resource requirements for patients under the influence of alcohol in the emergency department.

Resource	*n*	%
Radiology	2186	95.46
Laboratory tests	1895	82.75
Monitoring	1396	60.96
Medications	1390	60.69
Sedation	502	21.92
Suturing	159	6.94
Decontamination	25	1.09

**Table 4 healthcare-11-00786-t004:** Most common diagnoses in patients under the influence of alcohol in the emergency department in ICD-10 coding system.

Diagnosis (ICD-10 Code)	*n*	%
Alcohol-related disorders (F10)	592	25.82
Toxic effect of alcohol (T51)	359	15.66
Superficial injury of head (S00)	122	5.32
Open wound of head (S01)	118	5.15
Abdominal and pelvic pain (R10)	63	2.75
Pain in throat and chest (R07)	58	2.53
Evidence of alcohol involvement determined by level of intoxication (Y91)	53	2.31
Other general symptoms and signs (R68)	41	1.79
Syncope and collapse (R55)	37	1.61
Epilepsy and recurrent seizures (G40)	32	1.40
Intracranial injury (S06)	27	1.17
Other and unspecified injuries of head (S09)	24	1.04

**Table 5 healthcare-11-00786-t005:** Localizations and special characteristics of sustained traumatic injuries in patients under the influence of alcohol in the emergency department. Trauma in anamnesis refers to trauma reported by a patient, but not noticed by a physician during physical examination.

Localization/Characteristics of Injury	*n*	%
Trauma in anamnesis	610	26.64
Injuries in examination	451	19.69
Injuries to the head	344	76.27
Injuries to the neck	4	0.89
Injuries to the thorax	13	2.88
Injuries to the abdomen, lower back, lumbar spine, and pelvis	14	3.10
Injuries to the lower extremity	22	4.88
Injuries to the upper extremity	25	5.54
Injuries involving multiple or unspecified body regions and foreign bodies	24	5.32
Burns, corrosions and frostbites	4	0.89

## Data Availability

Data are unavailable to be provided due to privacy.

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
