# Peer review of "The Burden of Alcohol-Related Emergency Department Visits in a Hospital of a Large European City"

_healthcare, 2023, doi:10.3390/healthcare11060786_

Round 1
Reviewer 1 Report
Dear authors,
Thank you for your interesting paper.
Please
In the text where you mentioned BAC first, add its definition.
I suggest adding the city and country to the title
The female percent on line 121 can be deleted (since you mentioned the male percent previously).
4 - Please review the results section and include all table numbers related to the results.
Please add "body" before alcohol concentration on line 126 or use its abbreviation (BAC).
6- In line 126, please add BAC after lowest.
7-1- Please review all the text and check all abbreviations, number of references, and tables.
Author Response
Dear reviewer,
thank you for your time and effort that you dedicated to improve our manuscript. We’ve incorporated your suggestions. We have highlighted the changes within the manuscript.
Here is our point-by-point response to the comments and concerns.
Comment 1: In the text where you mentioned BAC first, add its definition.
Response: Thank you for pointing this out. We agree with this and have incorporated your suggestion.
Comment 2: I suggest adding the city and country to the title
Response: Thank you for pointing this out. We decided to change the title to “The burden of alcohol-related emergency department visits in a hospital of a large European city.”, which was recommended by one of the reviewers.
Comment 3: The female percent on line 121 can be deleted (since you mentioned the male percent previously).
Response: Thank you for pointing this out. We agree with this and have incorporated your suggestion.
Comment 4: Please review the results section and include all table numbers related to the results.
Response: Thank you for pointing this out. All four tables related to the results are mentioned and included in the results section.
Comment 5: Please add "body" before alcohol concentration on line 126 or use its abbreviation (BAC).
Response: Thank you for pointing this out. We agree with this and have incorporated your suggestion.
Comment 6: In line 126, please add BAC after lowest.
Response: Thank you for pointing this out. We agree with this and have incorporated your suggestion.
Comment 7: Please review all the text and check all abbreviations, number of references, and tables.
Response: Thank you for this suggestion. Following your recommendation we reviewed all the text and checked all abbreviations, number of references, and tables throughout the manuscript.
We look forward to hearing from you in due time regarding our submission and to respond to any further questions and comments you may have.
Sincerely,
Hanna Cholerzynska
Reviewer 2 Report
This article is retrospective, and descriptive, but interesting. It describes the number and characteristics of patients seen in the emergency department in a single hospital in Poland, which makes its value limited.
1.-There are some unclear abbreviations. For example, BAC that appears for the first time in the abstract.
2.-The authors include patients 18 years of age or older. However, it would have been interesting to see the number of adolescent patients who consult with a problem due to alcohol consumption.
3.-884 records were excluded due to a lack of sufficient data. It is almost 28%. This is a high ratio. The authors should explain how this may influence the results and conclusions. This high rate of exclusions is a significant limitation.
4.-The title of table 3 is not correct since it is the same as in table 2
5.- The first three lines of the discussion are repeated from the last paragraph of the results. Please, correct this.
6.- On the other hand, the title says "in a large European city," but the study was carried out only in one hospital in the city, so it would be better to say "in a hospital of a large European city."
Author Response
Dear reviewer,
thank you for your time and effort that you dedicated to improve our manuscript. We’ve incorporated your suggestions. We have highlighted the changes within the manuscript.
Here is our point-by-point response to the comments and concerns.
Comment 1: There are some unclear abbreviations. For example, BAC that appears for the first time in the abstract.
Response: Thank you for pointing this out. We agree with this and have incorporated your suggestion.
Comment 2: The authors include patients 18 years of age or older. However, it would have been interesting to see the number of adolescent patients who consult with a problem due to alcohol consumption.
Response: Thank you for this suggestion. We agree that it would have been interesting to explore this aspect. However, our hospital is serving the adult population and usually no children present to the ED, as the children’s hospital has a pediatric emergency department. Moreover, we checked and no patients under 18 that would fit into our study group presented to the ED.
Comment 3: 884 records were excluded due to a lack of sufficient data. It is almost 28%. This is a high ratio. The authors should explain how this may influence the results and conclusions. This high rate of exclusions is a significant limitation.
Response: Agree. We have, accordingly, modified the first paragraph of the result to explain why those patients were excluded and why it did not heavily affect our results. Also, in the last paragraph of the discussion we added information about this limitation to the study.
Comment 4: The title of table 3 is not correct since it is the same as in table 2
Response: Thank you for pointing this out. We agree with this and have incorporated your suggestion.
Comment 5: The first three lines of the discussion are repeated from the last paragraph of the results. Please, correct this.
Response: Thank you for pointing this out. We agree with this and have incorporated your suggestion.
Comment 6: On the other hand, the title says "in a large European city," but the study was carried out only in one hospital in the city, so it would be better to say "in a hospital of a large European city."
Response: Thank you for pointing this out. We agree with this and have incorporated your suggestion.
We look forward to hearing from you in due time regarding our submission and to respond to any further questions and comments you may have.
Sincerely,
Hanna Cholerzynska
Reviewer 3 Report
1. in the last paragraph of introduction, authors should clearly state what type of burden that authors want to investigate similar in the method (LOS and resource needed)
2. in this paper, authors did not use at least a multivariate analysis to determine the impact of alcohol related ED attendances to ED burden. the result has a low internal validity due to not controlling confounding or other factors. Please re-analyse using the most appropriate multivariate analysis according to your outcome such as linear regression for LOS or authors can use logistic regression if authors choose to categorized the outcome into two category.
3. in the discussion, authors mentioned that there is an inverse association between BAC and admission. Authors also said that it could be related to the association between lower BAC with other comorbidities or health condition. This assumption could be resolved by analyzing the rate of BAC with the presence of comorbidities or other health conditions among participants.
4. Please re-write your table, the first table should be about the characteristics of participants and the second table could mention about bivariate analysis result and the third one is about multivariate analysis result.
Author Response
Dear reviewer,
thank you for your time and effort that you dedicated to improve our manuscript. We’ve incorporated your suggestions. We have highlighted the changes within the manuscript.
Here is our point-by-point response to the comments and concerns.
Comment 1: in the last paragraph of introduction, authors should clearly state what type of burden that authors want to investigate similar in the method (LOS and resource needed)
Response: Thank you for this suggestion. We have, accordingly, modified the last paragraph of introduction to emphasize what type of burden we want to investigate.
Comment 2: in this paper, authors did not use at least a multivariate analysis to determine the impact of alcohol related ED attendances to ED burden. the result has a low internal validity due to not controlling confounding or other factors. Please re-analyse using the most appropriate multivariate analysis according to your outcome such as linear regression for LOS or authors can use logistic regression if authors choose to categorized the outcome into two category.
Response: Thank you for this suggestion. We agree with this comment. Therefore, we have added appropriate fragments to the statistical analysis in methods, results and discussion sections.
Comment 3: in the discussion, authors mentioned that there is an inverse association between BAC and admission. Authors also said that it could be related to the association between lower BAC with other comorbidities or health condition. This assumption could be resolved by analyzing the rate of BAC with the presence of comorbidities or other health conditions among participants.
Response: Thank you for this suggestion. It would have been interesting to explore this aspect. However, in the case of our study, it was impossible to achieve. This is a retrospective study and we did not implement a data collecting protocol, so some data would be missing, thus the correlation would be unreliable.
Comment 4: Please re-write your table, the first table should be about the characteristics of participants and the second table could mention about bivariate analysis result and the third one is about multivariate analysis result.
Response: Thank you for this suggestion. It would have been interesting to present our data in this way. However, we agreed that our initial idea for tables’ structure better fits the scope of the study and demonstrates data that we would like to emphasise.
We look forward to hearing from you in due time regarding our submission and to respond to any further questions and comments you may have.
Sincerely,
Hanna Cholerzynska
Reviewer 4 Report
This is an interesting retrospective study on alcohol-related presentations to a large ED in Poland. Although these findings are specific to Poland, I think similar issues can be recognized elsewhere in Europe and the US.
Minor commment:
Table 4- what is trauma in anamnesis?
Author Response
Dear reviewer,
thank you for your time and effort that you dedicated to improve our manuscript. We’ve incorporated your suggestions. We have highlighted the changes within the manuscript.
Here is our point-by-point response to the comments and concerns.
Comment 1: Table 4- what is trauma in anamnesis?
Response: This refers to trauma reported by a patient but not noticed by a physician during physical examination. We have added clarification in the table’s caption.
We look forward to hearing from you in due time regarding our submission and to respond to any further questions and comments you may have.
Sincerely,
Hanna Cholerzynska
Reviewer 5 Report
Abstract
Mention the cities which are studied in the abstract.
Include the sample size and study period.
Include the significance and results of the study in a brief manner.
Introduction
Line 55-57: The explanation of EDs could be more concise and to-the-point, as it seems somewhat repetitive.
Line 63-64: The purpose of the study could be more clearly stated, as the sentences seem somewhat disconnected.
Methods
The methodology seems to be described in enough detail and the process of data collection, primary and secondary outcomes, and statistical analysis appear to be explained thoroughly. However, the author should rework this section and follow the STROBE checklist for obesrvational study.
Result
the table and figure could be improved for better clarity and accessibility to the readers. For example, adding captions to the table and figure, using appropriate units and scales, and adding a brief explanation of the data sources.
Discussion
No major insufficiencies were found in the Discussion and Conclusion section of the paper. However, some areas for improvement are as follows:
-Line 225: It would be helpful to provide more details about the multi-centered study by Klein et al. that found no correlation between BAC and LOS.
-Line 234: It would be helpful to mention any guidelines that exist in other countries or internationally.
-Line 235: More information on what constitutes "good clinical practice" would be useful.
-Line 237-239: The explanation could be further expanded and clarified, for example by providing more detail about what symptoms of alcohol intoxication may require hospitalization.
Author Response
Dear reviewer,
thank you for your time and effort that you dedicated to improve our manuscript. We’ve incorporated your suggestions. We have highlighted the changes within the manuscript.
Here is our point-by-point response to the comments and concerns.
Comment 1: Mention the cities which are studied in the abstract.
Response: Thank you for pointing this out. We agree with this and have incorporated your suggestion.
Comment 2: Include the sample size and study period.
Response: Thank you for pointing this out. We agree with this and have incorporated your suggestion.
Comment 3: Include the significance and results of the study in a brief manner.
Response: Agree. We have, accordingly, modified the Abstract to express significance and results of the study in a brief manner.
Comment 4: Line 55-57: The explanation of EDs could be more concise and to-the-point, as it seems somewhat repetitive.
Response: Thank you for pointing this out. We agree with this comment. Therefore, we have modified the description of EDs to be more concise.
Comment 5: Line 63-64: The purpose of the study could be more clearly stated, as the sentences seem somewhat disconnected.
Response: Thank you for pointing this out. We agree with this and have incorporated your suggestion.
Comment 6: The methodology seems to be described in enough detail and the process of data collection, primary and secondary outcomes, and statistical analysis appear to be explained thoroughly. However, the author should rework this section and follow the STROBE checklist for obesrvational study.
Response: Thank you for pointing this out. We agree with this and have incorporated your suggestion.
Comment 7: the table and figure could be improved for better clarity and accessibility to the readers. For example, adding captions to the table and figure, using appropriate units and scales, and adding a brief explanation of the data sources.
Response: Agree. We have, accordingly, modified the captions to to improve their clarity and accessibility to the readers.
Comment 8: Line 225: It would be helpful to provide more details about the multi-centered study by Klein et al. that found no correlation between BAC and LOS.
Response: Thank you for pointing this out. We agree with this and have incorporated your suggestion.
Comment 9: Line 234: It would be helpful to mention any guidelines that exist in other countries or internationally.
Response: Thank you for this suggestion. It would be helpful however, we did not find any official guidelines.
Comment 10: Line 235: More information on what constitutes "good clinical practice" would be useful.
Response: Thank you for pointing this out. We decided to delete the word “good” and leave just clinical practice, as it is not an official guideline or recommendation issued by a scientific committee or government. This phrase (and sentence) represents the way of management that is practiced in polish hospitals.
Comment 11: Line 237-239: The explanation could be further expanded and clarified, for example by providing more detail about what symptoms of alcohol intoxication may require hospitalization.
Response: Thank you for pointing this out. We agree with this and therefore added appropriate information in the last paragraph on page 8.
We look forward to hearing from you in due time regarding our submission and to respond to any further questions and comments you may have.
Sincerely,
Hanna Cholerzynska
Round 2
Reviewer 2 Report
The authors have responded to my requirements about their manuscript, and I think it can be published.
Author Response
Dear reviewer,
thank you for your time and effort that you dedicated to improve our manuscript.
Reviewer 3 Report
almost all of the queries has been answered by authors. There is one more query need to be resolved. Please add a multivariate analysis table and not just one sentence because the most important to answer all of that is the multivariate analysis. We can't say which one get highest burden due to BAC if authors did not conduct and present proper multivariate analysis result and table. the multivariate should include BAC as main independent and other variable as covariates such as age, gender, time, days, any laboratory or surgical intervention, other comorbidities. Without proper multivariate analysis, authors can only say describing about the burden of BAC not determining the impact of BAC.
Author Response
Dear reviewer,
thank you for your time and effort that you dedicated to improve our manuscript. We’ve incorporated your suggestions.
Comment 1: almost all of the queries has been answered by authors. There is one more query need to be resolved. Please add a multivariate analysis table and not just one sentence because the most important to answer all of that is the multivariate analysis. We can't say which one get highest burden due to BAC if authors did not conduct and present proper multivariate analysis result and table. the multivariate should include BAC as main independent and other variable as covariates such as age, gender, time, days, any laboratory or surgical intervention, other comorbidities. Without proper multivariate analysis, authors can only say describing about the burden of BAC not determining the impact of BAC.
Response: Thank you for pointing this out. We agree with this and therefore we added appropriate parts of the paper in sections: material and methods (end of point 2.7), results (end of 3.1.2) and discussion (page 10).